# Use of the Sol–Gel Method for the Preparation of Coatings of Titanium Substrates with Hydroxyapatite for Biomedical Application

**Michelina Catauro** [1],*[ID], **Federico Barrino** [1][ID], **Ignazio Blanco** [2][ID], **Simona Piccolella** [3][ID] and **Severina Pacifico** [3][ID]

1 Department of Engineering, University of Campania "Luigi Vanvitelli", Via Roma 29, I-813031 Aversa, Italy; federico.barrino@unicampania.it
2 Department of Civil Engineering and Architecture and UdR-Catania Consorzio INSTM, University of Catania, Viale Andrea Doria 6, 95125 Catania, Italy; iblanco@unict.it
3 Department of Environmental, Biological and Pharmaceutical Sciences and Technologies, University of Campania "Luigi Vanvitelli", Via Vivaldi 43, 81100 Caserta, Italy; simona.piccolella@unicampania.it (S.P.); severina.pacifico@unicampania.it (S.P.)
* Correspondence: michelina.catauro@unicampania.it

**Abstract:** Hydroxyapatite (HA) was coated onto the surface of commercially pure titanium grade 4 (a material generally used for implant application) by a dip coating method using HA sol. Hydroxyapatite sol was synthesized via sol–gel using $Ca(NO_3)_2 \cdot 4H_2O$ and $P_2O_5$ as precursors. The surface of the HA coating was homogeneous, as determined by scanning electron microscopy (SEM), attenuated total reflectance Fourier transform infrared (ATR-FTIR), and X-ray diffraction (XRD), which allowed the materials to be characterized. The bioactivity of the synthesized materials and their efficiency for use as future bone implants was confirmed by observing the formation of a layer of hydroxyapatite on the surface of the samples soaked in a fluid simulating the composition of human blood plasma. To verify the biocompatibility of the obtained biomaterial, fibroblasts were grown on a glass surface and were tested for viability after 24 h. The results of the WST-8 analysis suggest that the HA systems, prepared by the sol–gel method, are most suitable for modifying the surface of titanium implants and improving their biocompatibility.

**Keywords:** sol–gel synthesis; hydroxyapatite; titanium substrate coating; biocompatibility

## 1. Introduction

The aim of tissue engineering is to develop substitutes, organic or synthetic, which are able to restore or maintain the function of damaged tissue within the human body. This science is the synergistic union between life sciences and engineering [1]. Tissue engineering should ideally produce substitute tissue that can "grow" with the patient, overcoming the limitations of conventional treatments that are based on biomaterial implant and organ transplantation [2].

Scaffolds must meet requirements, such as biocompatibility, surface topography, and chemistry, in order to promote cell adhesion, proliferation and differentiation, controlled biodegradability, and bioactivity [3]. Taking into account these fundamental characteristics, different types of materials (natural, synthetic, semisynthetic, and hybrids) have been synthesized to develop scaffolds according to the specific tissue that requires regeneration, with focused attention on the concept of controlled release of biological molecules or cell differentiation induction [4,5]. Scaffolds for bone tissue engineering must have suitable mechanical properties. For this reason, ceramic materials such as hydroxyapatite, silica, zirconia, and tricalcium phosphate have been considered. These materials have proven to be

biocompatible and bioactive, which are basic characteristics necessary in the orthopedic or dental fields [6,7].

Hydroxyapatite (HA) has attracted much attention as a material for artificial bones, scaffolds for tissue engineering, and chromatographic packaging [8]. Due to the chemical similarity between HA and the mineralized bone of human tissue, synthetic HA exhibits strong affinity to host hard tissues and is able to form chemical bonds with the host tissue. Metallic implants, on the other hand, do not interact with the biological environment. However, metallic alloy is a bioinert material [9,10]. In recent years, coatings that can be applied to metal alloys have been developed. These have been used in the biomedical field to improve interaction with hard tissue [11,12]. Nanocrystalline HA is synthesized by various processes, such as hydrothermal [13], mechanochemical [14], precipitation [15], hydrolysis [16], and sol–gel methods [17]. The sol–gel method is able to improve the HA chemical homogeneity in comparison with other techniques, because it involves a mixing at the molecular level of calcium and phosphorus precursors. This technique can also form thin film coatings with a simple process such as dip coating.

The dip coating technique allows substrates of different materials to be coated with complex shapes [18–23].

In the present work, synthetic HA was synthesized by using the sol–gel method. Hydroxyapatite sol was used to coat metallic substrate titanium grade 4 (Ti-4), which is widely used in implants for orthopedic, dental, and orthodontic wires. The dip coating technique was used to make a thin layer on substrates. Scanning electron microscope (SEM), attenuated total reflectance Fourier transform infrared (ATR-FTIR), and X-ray diffraction (XRD) were then used for chemical and morphological coating characterizations. The objective of the coatings is to combine the favorable mechanical properties of titanium with the excellent biological properties of HA. For that reason, after coating, bioactivity and biocompatibility of the coated substrates were preliminarily evaluated and compared with those of uncoated substrates. Bioactivity tests were performed by soaking samples in simulated body fluids (SBF) [24]. Biocompatible properties were assessed through WST-8 tests on NIH-3T3 murine fibroblasts.

## 2. Materials and Methods

### 2.1. Sol–Gel Synthesis

Hydroxyapatite for coating was synthesized using two different chemical reagents as precursors. Phosphorous pentoxide (0.006 mol) ($P_2O_5$, Sigma Aldrich, St. Louis, MI, USA) and calcium nitrate tetrahydrate (0.02 mol) ($Ca(NO_3)_2 \cdot 4H_2O$, Sigma Aldrich) were dissolved in absolute ethanol (EtOH, 99.8% Sigma-Aldrich) to form 1 M and 2.5 M solutions respectively. $NH_4OH$ was utilized to adjust the pH to 11. The molar ratio of precursors Ca/P was 1.67, which is the desired ratio observed in hydroxyapatite. The flow chart of the synthesis is depicted in Figure 1. A $P_2O_5$ and ethanol solution was stirred for 24 h to obtain a clear solution that was added dropwise to the stirred calcium nitrate solution. Afterwards, the solution was stirred slowly for 2 h and left at room temperature, protected from dust for 24 h.

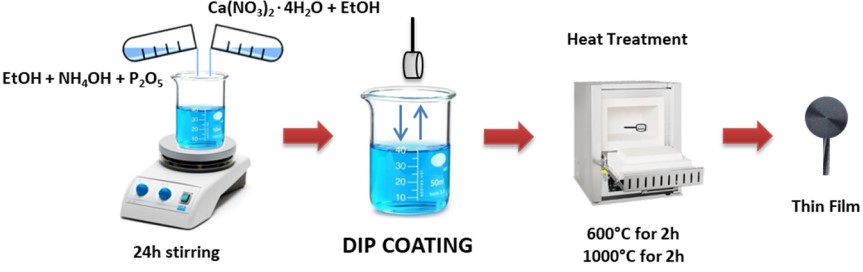

**Figure 1.** Schematic flow process chart for the synthesis of hydroxyapatite (HA).

### 2.2. Coating Procedure

Hydroxyapatite sol was used to coat commercially pure titanium grade 4 disks with 8 mm diameter (Sweden & Martina, Padua, Italy). A KSV LM dip coater was used for depositing sol–gel coatings. Before coating, the substrates were washed with acetone and then subjected to a passivation process using $HNO_3$ 65 wt%. The coating procedure was carried out 24 h after synthesis, and a withdrawal speed equal to 15 mm/min was used. The coated substrates were heated at 600 and 1000 °C for 2 h.

### 2.3. Chemical and Morphological Characterization

The chemical nature of the hybrid materials (600 °C untreated and 1000 °C treated) was investigated by X-ray diffraction analysis using a Philips diffractometer (Philips Electronic Instruments Co., Model PW 1730, Eindoven, Netherlands). Powder samples were scanned from $2\Theta = 10$ to $80°$ using CuK$\alpha$ radiation. Both heated and non-heat-treated HA samples were characterized chemically by ATR-FTIR and recorded on a Prestige-21 FTIR spectrometer equipped with an AIM-8800 infrared microscope (Shimazu, Tokyo, Japan), using the incorporated 3 mm diameter Ge ATR semicircular prism. Spectra were acquired in the 4000–650 $cm^{-1}$ range, using 45 scans and 8 $cm^{-1}$ resolution. Spectra analyses were completed using Prestige software (IRsolution, version 1.10). Coating morphologies were investigated by a scanning electron microscope (SEM) Quanta 200 FEI (Europe Company, Eindhoven, Netherlands) fitted with energy dispersive X-ray spectroscopy (EDS). Samples were fixed on an aluminum stub with colloidal graphite and gold metalized using a K550X Sputter Coater (Emitech, Ashford, UK).

### 2.4. In Vitro Bioactivity

The potential bone-bonding ability of a material when implanted in vivo can be predicted by the apatite-forming-ability test in vitro, in accordance with Kokubo et al. [24]. The test was carried out by soaking the titanium substrates, uncoated and coated, with HA after heating at 600 and 1000 °C for 7, 14, and 21 days, in a simulated body fluid (SBF). This is a solution prepared by adding several salts to distilled water in order to give a concentration of ions similar to those of human blood plasma. As the ratio between the total surface area exposed to the solution and its volume affects the formation of the HA layer, it was kept equal to 10. A water bath was used to maintain the polystyrene bottles, where the samples in the solution were placed at 37 °C. In order to avoid the depletion of ionic species in the SBF solution due to biomineral nucleation, the solution was replaced every 2 days.

SEM/EDX analysis was performed after each soaking time in order to evaluate the formation of a hydroxyapatite layer on the surface of the air-dried samples.

### 2.5. WST-8 Assay

To evaluate the coatings' biocompatibility, NIH-3T3 murine fibroblasts (ATCC, Manassas, Virginia, USA) were seeded on Ti-4 coated and uncoated substrates. Cell viability was tested via WST-8 assay (coated and uncoated substrates. Cell viability was tested via WST-8 assay WST-8 [2-(2-methoxy-4-nitrophenyl) -3-(4-nitrophenyl)-5-(2,4-disulfophenyl)-2H-tetrazolium, monosodium salt], which is water-soluble and purple in color, can be converted by mitochondrial dehydrogenases in living cells into yellow formazan soluble in culture medium. Thus, the formazan amount is directly proportional to the living cells number. NIH-3T3 cells were grown in a humified chamber at 37 °C, with 5% $CO_2$. DMEM medium (Gibco, Carlsbad, CA, USA) was supplemented with 10% (v/v) fetal bovine serum and 1% penicillin/streptomycin. NIH-3T3 cells were seeded on the surface of coated and uncoated Ti-4 disks, previously placed on the bottom of a 24-well plate. Cell density was equal to $5 \times 10^3$. After 24 h exposure time, a WST-8 viability test was carried out. The absorbance was measured at 450 nm using a Victor3™ multilabel plate reader (PerkinElmer, Shelton, CT, USA). Cells grown on uncoated Ti-4 disks were defined as having 100% vitality. Cells grown on Ti-4 coated and uncoated substrates were observed by an inverted phase contrast and brightfield Zeiss Primo Vert Microscope (Jena, Germany), and representative images were acquired without specific staining.

## 3. Results

### 3.1. Spectroscopic and Structural Characterization

The XRD results of sol–gel untreated (a), treated at 600 °C (b), and treated at 1000 °C (c) are shown in Figure 2. The sintering temperature plays an important role in HA formation. In fact, increasing the sintering temperature causes the samples to show several distinct and narrow peaks, which suggests that there is an increase in crystallite size.

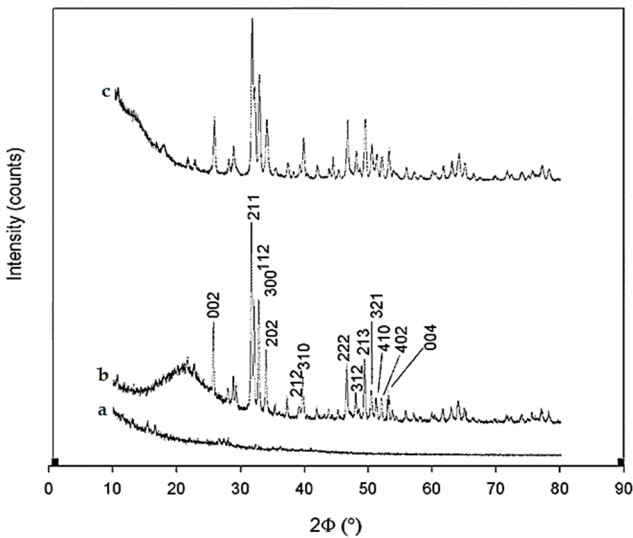

**Figure 2.** X-ray diffraction (XRD) results of HA sol–gel (**a**) untreated, (**b**) treated at 600 °C, and (**c**) treated at 1000 °C.

In spectra of heat-treated samples, typical peak positions corresponding to HA reflections were observed, including those with higher intensity as well as the peak positions ($2\Phi$) at 27.76, 31.80, 32.26, 33, and 53.06, which were in line with the (002), (211), (112), (300), and (004) reflections of HA [25].

ATR-FTIR spectra of the hydroxyapatite coating untreated (a), treated at 600 °C (b), and treated at 1000 °C (c) are reported in Figure 3.

In spectra of untreated samples, strong signals due to the asymmetric and symmetric stretching of nitrate ions at 1423, 1382, and 1354 $cm^{-1}$ were detectable [26]. The bands at 3420 and 1650 $cm^{-1}$ were attributable to water used in synthesis. Furthermore, sharp peaks at 1047 and 823 $cm^{-1}$ could be ascribed to the bending modes of the nitrate ions of $Ca(NO_3)_2 \cdot 4H_2O$. Indeed, only when the thermal degradation of nitrates occurs (at a temperature beyond 350 °C) do calcium ions enter into the network by diffusion. Before such temperature, calcium nitrate salt coats the material network; therefore, the FTIR nitrate signals appear strong.

The presence of the asymmetric and symmetric stretching of P-O bonds at 1090 and 960 $cm^{-1}$, however, proves that a HA network was formed [27,28]. After heat treatment at 600 °C, another peak relating to the vibrational mode of hydroxyapatite P–O groups (1033 $cm^{-1}$) appeared (Figure 3b), whereas HA hydroxyl ions occurred at wavenumbers higher than 3420 $cm^{-1}$. The presence of peaks typical of $CO_3^{2-}$ vibrations at 1450 and 873 $cm^{-1}$ resulted from carbonated hydroxyapatite.

The HA spectrum of coating treated at 1000 °C is reported in Figure 3c. The intensities of peaks at 1450 and 873 $cm^{-1}$ were evident, reflecting $CO_3^{2-}$ released as volatile gas at higher calcination temperatures, while the band at 3570 $cm^{-1}$ disappeared. Finally, a red-shift from 1033 to 1040 $cm^{-1}$ was attributable to P–O vibrational mode, because, according to XRD, peaks become narrower at this calcination temperature.

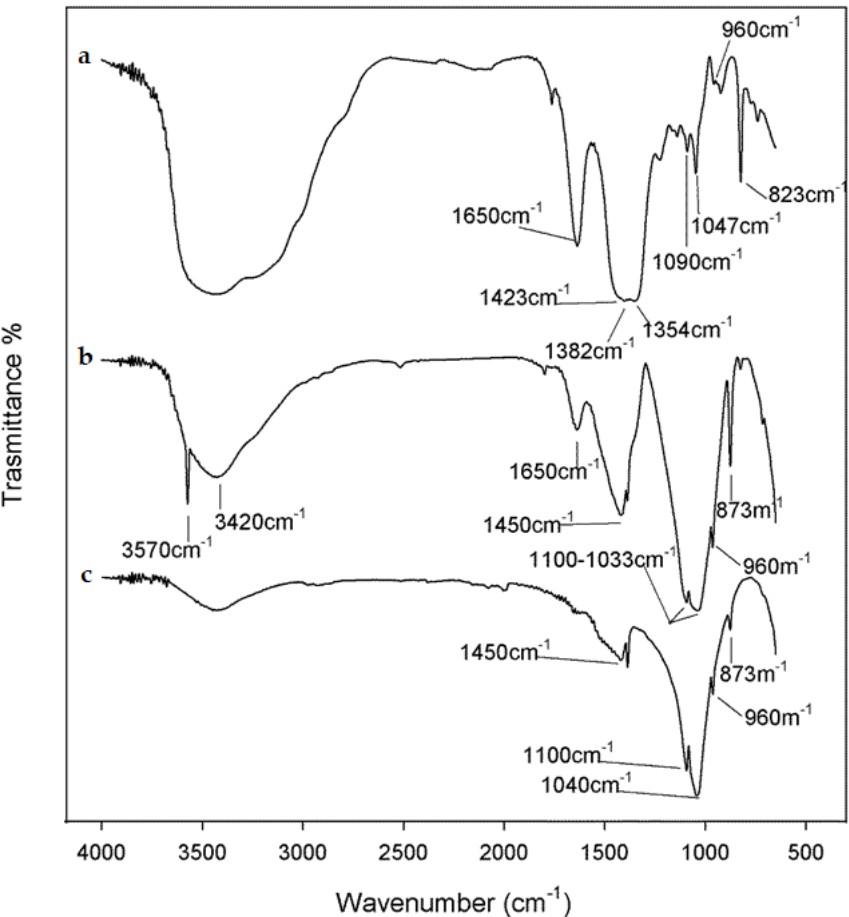

**Figure 3.** ATR-FTIR spectra of HA layers after (**a**) untreated, (**b**) treated at 600 °C, and (**c**) treated at 1000 °C.

### 3.2. Bioactivity Test

SEM images of HA coatings heat treated at 600 °C and 1000 °C are shown in Figure 4. Both samples appeared to have a porous structure with micropores of 3–4 µm in diameter on the surface. During sample calcination, a partial HA carbonation occurs due to atmospheric $CO_2$ as a consequence of high calcium content in the prepared samples. This phenomenon is responsible for the Ca/P ratio of 1.50.

Sol–gel hydroxyapatite coating after soaking in SBF for 7, 14, and 21 days (Figure 5) showed the globular formations typical of physiological hydroxyapatite. Indeed, this was more evident at 21 days than at 7 and 14 days. The Ca/P ratio was 1.30 for the additional presence of carbonates in simulated body fluid solution.

### 3.3. Cytotoxicity Assessment

Cell viability assay highlighted that HA coatings were biocompatible, with rates higher than uncoated substrates (Figure 6A). The latter were considered to be 100% viable. Thermal treatments differed slightly from each other; it clearly appeared that HA coating favored attachment and proliferation, although cells were likely found in clusters (Figure 6B).

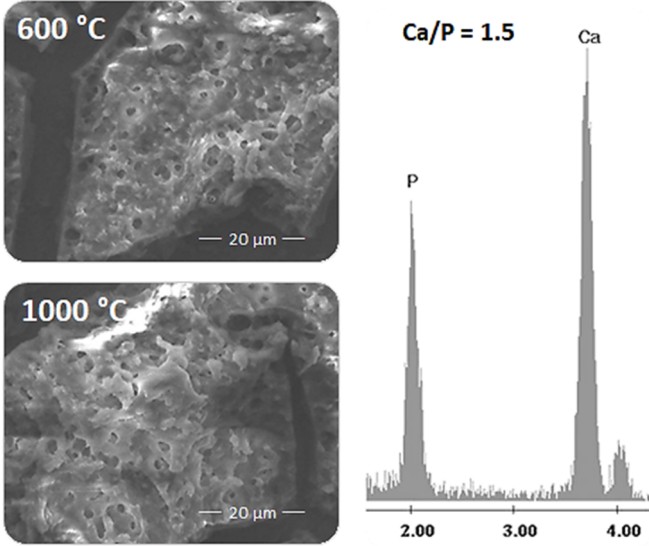

**Figure 4.** SEM-EDS of hydroxyapatite coating after heat treatment at 600 °C and 1000 °C.

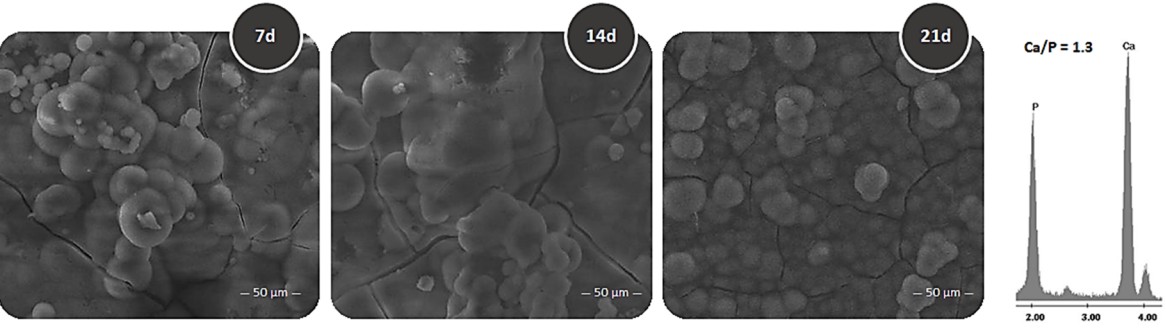

**Figure 5.** SEM-EDS of hydroxyapatite coating after soaking in simulated bodily fluids (SBF) for 7, 14, and 21 days.

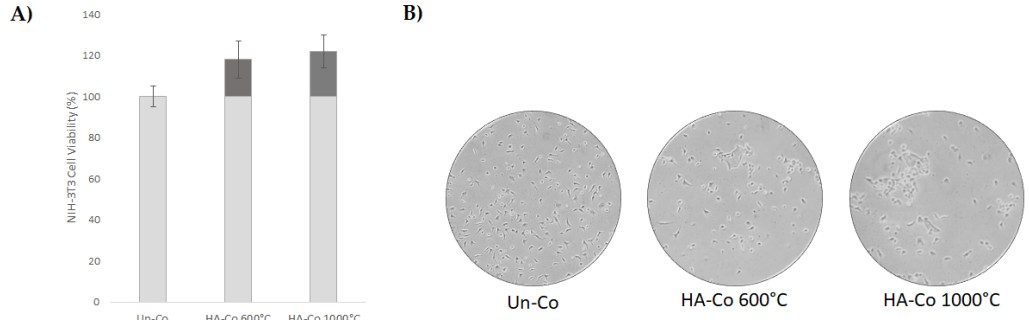

**Figure 6.** (**A**) Results of WST-8 assay expressed as % of viability of cells seeded on hydroxyapatite heat treated at 600 °C and 1000 °C. (**B**) Cell morphology images of NIH-3T3 cells seeded on hydroxyapatite heat treated at 600 °C and 1000 °C were acquired by Inverted Phase Contrast Brightfield Zeiss Primo Vert Microscope and representatively shown. Un-Co = uncoated; HA-Co = hydroxyapatite coated.

## 4. Conclusions

HA was synthesized using the sol–gel method and then used for coating titanium disks. The substrates were coated with the dip coating procedure, and an improvement in biocompatibility was demonstrated. FT-IR analysis of untreated, 600 °C-, and 1000 °C-treated samples indicated that heat

treatment degraded the nitrates used in synthesis, which may adversely affect the biological response to the material. This treatment also showed a carbonation of HA at high temperatures. SEM analysis showed the film morphology of various heat treatments. Microporous structure could promote cell adhesion and proliferation. Furthermore, coated samples appeared more bioactive and biocompatible than uncoated ones.

**Author Contributions:** Validation, M.C.; formal analysis, F.B., S.P. (Simona Piccolella), and S.P. (Severina Pacifico); data curation, M.C. and I.B.; writing—original draft preparation, M.C.; supervision, M.C. All authors have read and agreed to the published version of the manuscript.

**Funding:** The work was financially supported by V:ALERE 2019 grant support from Università degli studi della Campania "L. Vanvitelli" of CHIMERA.

**Conflicts of Interest:** The authors declare no conflicts of interest.

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
