# Peer review of "Use of the Sol–Gel Method for the Preparation of Coatings of Titanium Substrates with Hydroxyapatite for Biomedical Application"

_coatings, doi:10.3390/coatings10030203_

Round 1

Reviewer 1 Report

Manuscript COATINGS-693376

Comment: The manuscript “Coatings of Titanium Substrates with Hydroxyapatite Prepared by the Sol-Gel Method for Biomedical Application” is focused on new coatings of pure titanium disks using Hydroxyapatite systems, prepared by sol-gel method. HA homogenous coatings obtained at different temperature were well characterized using scanning electron microscopy (SEM), ATR-FTIR and X-Ray diffraction, comparing these results with those obtained considering untreated surfaces. The hydroxyapatite layer on the treated titanium surfaces is observed. Then the bioactivity of these materials is investigated: The detected microstructure of titanium surface coated with HA prepared by sol-gel method shows micropores of suitable dimensions and turns out to be more able to promote cell adhesion and osseointegration, so important for biomaterials for biomedical applications.

The manuscript is interesting, well written and well organized. I think this manuscript may contribute to have coatings of titanium surface for biomedical applications, in particular to enhance the osseointegration process and in general the biocompatibility in contact with biological environment.

Therefore, I recommend this paper for publication in the present version.

Author Response

REVIEWER 1

Comment: The manuscript “Coatings of Titanium Substrates with Hydroxyapatite Prepared by the Sol-Gel Method for Biomedical Application” is focused on new coatings of pure titanium disks using Hydroxyapatite systems, prepared by sol-gel method. HA homogenous coatings obtained at different temperature were well characterized using scanning electron microscopy (SEM), ATR-FTIR and X-Ray diffraction, comparing these results with those obtained considering untreated surfaces. The hydroxyapatite layer on the treated titanium surfaces is observed. Then the bioactivity of these materials is investigated: The detected microstructure of titanium surface coated with HA prepared by sol-gel method shows micropores of suitable dimensions and turns out to be more able to promote cell adhesion and osseointegration, so important for biomaterials for biomedical applications.

The manuscript is interesting, well written and well organized. I think this manuscript may contribute to have coatings of titanium surface for biomedical applications, in particular to enhance the osseointegration process and in general the biocompatibility in contact with biological environment.

Therefore, I recommend this paper for publication in the present version.

The authors are grateful to the reviewer for his/her positive comments.

Reviewer 2 Report

The authors present the obtaining of Hap coatings on Ti substrates.

The manuscript is not well organized and it lacks important characterization of the samples.

In the new presentation of the article we suggest the authors to do the following modifications:

They should present the XRD spectra on the layers obtained after the deposition, treated and untreated.

Also, they should present SEM images with better resolution for the layers, before and after thermal treatment.

Also, the EDS studies should also be added to the manuscript.

Moreover, AFM investigations should be performed on the coatings for assessing the surface morphology and roughness.   

As for biological studies, a study of the influence of the heat treatment applied to the layers on the morphology of cells should also be added to the manuscript.

Author Response

REVIEWER 2

Comment :The authors present the obtaining of Hap coatings on Ti substrates.

The manuscript is not well organized and it lacks important characterization of the samples.

In the new presentation of the article we suggest the authors to do the following modifications:

They should present the XRD spectra on the layers obtained after the deposition, treated and untreated.

Also, they should present SEM images with better resolution for the layers, before and after thermal treatment.

Also, the EDS studies should also be added to the manuscript.

Moreover, AFM investigations should be performed on the coatings for assessing the surface morphology and roughness.   As for biological studies, a study of the influence of the heat treatment applied to the layers on the morphology of cells should also be added to the manuscript.

The authors thank the reviewer for its comments and suggestions. The manuscript has been revised accordingly.

Data from EDS analysis have been added, and are now reported in Figures 4 and 5. The resolution of the SEM images has been improved. Cell morphology was only preliminarily observed in treated NIH-3T3 cells. Further studies are on-going and will be addressed to better clarify the influence of the heat treatment applied to the layers on the morphology of cells.

Reviewer 3 Report

The authors made the research entitled "Use of the sol-gel method for the preparation of coatings of titanium substrates with Hydroxyapatite for biomedical application". The reviewer suggests a major revision.

For the sol-gel route in the Introduction, several references were recommended:

Zhou, et al. Fabrication of 3D TiO2 micromesh on Silicon surface and its effects on platelet adhesion. Materials Letters, 2014, 132: 149-152.

For the cell test, there is only a semi quantitative statistical chart. The fluorescence images of the fibroblasts should be provided. As the authors statement, this research was developed for potential application of bone implants. Why didn’t the authors make experiments on osteoblasts or osteoclast?

Author Response

REVIEWER 3

The authors made the research entitled "Use of the sol-gel method for the preparation of coatings of titanium substrates with Hydroxyapatite for biomedical application". The reviewer suggests a major revision.

For the sol-gel route in the Introduction, several references were recommended:

Zhou, et al. Fabrication of 3D TiO2 micromesh on Silicon surface and its effects on platelet adhesion. Materials Letters, 2014, 132: 149-152. For the cell test, there is only a semi quantitative statistical chart. The fluorescence images of the fibroblasts should be provided. As the authors statement, this research was developed for potential application of bone implants. Why didn’t the authors make experiments on osteoblasts or osteoclast?

The authors thank the reviewer for its comments and suggestions.

As suggested, the papers have been included as valuable references in the introduction section and in the reference list. Accordingly, all the references’ numbers have been changed.

The manuscript focuses on chemical characterization of synthetic HA, whereas preliminary data on biocompatibility have been provided. To this purpose, NIH-3T3 cells, commonly used as cell model for assessing in vitro cytotoxicity of new biomaterials, were used. Further studies are on-going and will be addressed, based on these preliminary and promising responses, towards osteoblast or osteoclast cells.

Reviewer 4 Report

The manuscript is clear, well arranged, the experimental results are well presented.

However, it would be useful to include some other references on this topic:

- Roest, R et al. Adhesion of sol–gel derived hydroxyapatite nanocoatings on anodised pure titanium and titanium (Ti6Al4V) alloy substrates. Surf. Coat. Technol. 2011, 205, 3520–3529.

- U. Anjaneyulu et al. Preparation and characterisation of sol–gel-derived hydroxyapatite nanoparticles and its coatings on medical grade Ti-6Al-4V alloy for biomedical applications Materials Technology, Advanced Performance Materials Volume 32, p.800,  2017

And reviews:

- Review: Surface Modification of Biomedical Titanium Alloy: Micromorphology, Microstructure Evolution and Biomedical Applications Wei Liu et al. Coatings 2019, 9, 249

- Choi AH, Sol-gel production of bioactive nanocoatings for medical applications. Part II: current research and development. Nanomedicine. 2007;2:51–61.

It would be useful/better to include statements about the novelty and the contribution to the development of areas of current scientific interest.

Author Response

REVIEWER 4

Comment :The manuscript is clear, well arranged, the experimental results are well presented.

However, it would be useful to include some other references on this topic:

- Roest, R et al. Adhesion of sol–gel derived hydroxyapatite nanocoatings on anodised pure titanium and titanium (Ti6Al4V) alloy substrates. Surf. Coat. Technol. 2011, 205, 3520–3529.

- U. Anjaneyulu et al. Preparation and characterisation of sol–gel-derived hydroxyapatite nanoparticles and its coatings on medical grade Ti-6Al-4V alloy for biomedical applications Materials Technology, Advanced Performance Materials Volume 32, p.800,  2017

And reviews:

- Review: Surface Modification of Biomedical Titanium Alloy: Micromorphology, Microstructure Evolution and Biomedical Applications Wei Liu et al. Coatings 2019, 9, 249

- Choi AH, Sol-gel production of bioactive nanocoatings for medical applications. Part II: current research and development. Nanomedicine. 2007;2:51–61.

The authors thank the reviewer for its comments and suggestions.

As suggested, the papers have been included as valuable references in the introduction section and in the reference list. Accordingly, all the references’ numbers have been changed.

Round 2

Reviewer 3 Report

The reviewer agrees this paper for publlication.